# REDESIGNING THE CLASSIFICATION LAYER BY RANDOMIZING THE CLASS REPRESENTATION VECTORS

## ABSTRACT

Neural image classification models typically consist of two components. The first is an image encoder, which is responsible for encoding a given raw image into a representative vector. The second is the classification component, which is often implemented by projecting the representative vector onto target class vectors. The target class vectors, along with the rest of the model parameters, are estimated so as to minimize the loss function.

In this paper, we analyze how simple design choices for the classification layer affect the learning dynamics. We show that the standard cross-entropy training implicitly captures visual similarities between different classes, which might deteriorate accuracy or even prevents some models from converging. We propose to draw the class vectors randomly and set them as fixed during training, thus invalidating the visual similarities encoded in these vectors. We analyze the effects of keeping the class vectors fixed and show that it can increase the inter-class separability, intra-class compactness, and the overall model accuracy, while maintaining the robustness to image corruptions and the generalization of the learned concepts.

## 1 INTRODUCTION

Deep learning models achieved breakthroughs in classification tasks, allowing setting state-of-the-art results in various fields such as speech recognition (Chiu et al., 2018), natural language processing (Vaswani et al., 2017), and computer vision (Huang et al., 2017). In image classification task, the most common approach of training the models is as follows: first, a convolutional neural network (CNN) is used to extract a representative vector, denoted here as *image representation* vector (also known as the *feature vector*). Then, at the classification layer, this vector is projected onto a set of weight vectors of the different target classes to create the class scores, as depicted in Fig. 1. Last, a softmax function is applied to normalize the class scores. During training, the parameters of both the CNN and the classification layer are updated to minimize the cross-entropy loss. We refer to this procedure as the *dot-product maximization* approach since such training ends up maximizing the dot-product between the image representation vector and the target weight vector.

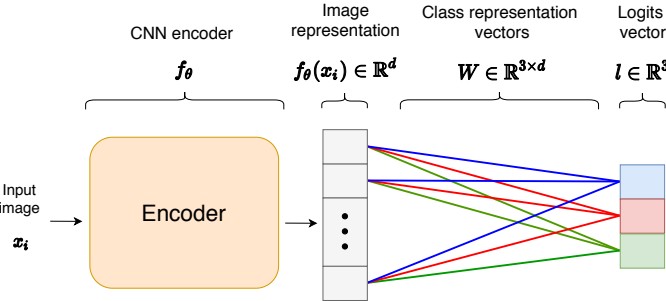

Figure 1: A scheme of an image classification model with three target classes. Edges from the same color compose a class representation vector.

Recently, it was demonstrated that despite the excellent performance of the dot-product maximization approach, it does not necessarily encourage discriminative learning of features, nor does it

enforce the intra-class compactness and inter-class separability (Liu et al., 2016; Wang et al., 2017; Liu et al., 2017). The intra-class compactness indicates how close image representations from the same class relate to each other, whereas the inter-class separability indicates how far away image representations from different classes are.

Several works have proposed different approaches to address these caveats (Liu et al., 2016; 2017; Wang et al., 2017; 2018b;a). One of the most effective yet most straightforward solutions that were proposed is *NormFace* (Wang et al., 2017), where it was suggested to maximize the cosine-similarity between vectors by normalizing both the image and class vectors. However, the authors found when minimizing the cosine-similarity directly, the models fail to converge, and hypothesized that the cause is due to the bounded range of the logits vector. To allow convergence, the authors added a scaling factor to multiply the logits vector. This approach has been widely adopted by multiple works (Wang et al., 2018b; Wojke & Bewley, 2018; Deng et al., 2019; Wang et al., 2018a; Fan et al., 2019). Here we will refer to this approach as the *cosine-similarity maximization* approach.

This paper is focused on redesigning the classification layer, and the its role while kept fixed during training. We show that the visual similarity between classes is implicitly captured by the class vectors when they are learned by maximizing either the *dot-product* or the *cosine-similarity* between the image representation vector and the class vectors. Then we show that the class vectors of visually similar categories are close in their angle in the space. We investigate the effects of excluding the class vectors from training and simply drawing them randomly distributed over a hypersphere. We demonstrate that this process, which eliminates the visual similarities from the classification layer, boosts accuracy, and improves the inter-class separability (using either *dot-product maximization* or *cosine-similarity maximization*). Moreover, we show that fixing the class representation vectors can solve the issues preventing from some cases to converge (under the *cosine-similarity* maximization approach), and can further increase the intra-class compactness. Last, we show that the generalization to the learned concepts and robustness to noise are both not influenced by ignoring the visual similarities encoded in the class vectors.

Recent work by Hoffer et al. (2018), suggested to fix the classification layer to allow increased computational and memory efficiencies. The authors showed that the performance of models with fixed classification layer are on par or slightly drop (up to 0.5% in absolute accuracy) when compared to models with non-fixed classification layer. However, this technique allows substantial reduction in the number of learned parameters. In the paper, the authors compared the performance of dot-product maximization models with a non-fixed classification layer against the performance of cosine-similarity maximization models with a fixed classification layer and integrated scaling factor. Such comparison might not indicate the benefits of fixing the classification layer, since the dot-product maximization is linear with respect to the image representation while the cosine-similarity maximization is not. On the other hand, in our paper, we compare fixed and non-fixed dot-product maximization models as well as fixed and non-fixed cosine-maximization models, and show that by fixing the classification layer the accuracy might boost by up to 4% in absolute accuracy. Moreover, while cosine-maximization models were suggested to improve the intra-class compactness, we reveal that by integrating a scaling factor to multiply the logits, the intra-class compactness is decreased. *We demonstrate that by fixing the classification layer in cosine-maximization models, the models can converge and achieve a high performance without the scaling factor, and significantly improve their intra-class compactness.*

The outline of this paper is as follows. In Sections 2 and 3, we formulate *dot-product* and *cosine-similarity maximization* models, respectively, and analyze the effects of fixing the class vectors. In Section 4, we describe the training procedure, compare the learning dynamics, and asses the generalization and robustness to corruptions of the evaluated models. We conclude the paper in Section 5.

## 2 FIXED DOT-PRODUCT MAXIMIZATION

Assume an image classification task with $m$ possible classes. Denote the training set of $N$ examples by $S = \{(x_i, y_i)\}_{i=1}^N$, where $x_i \in \mathcal{X}$ is the $i$-th instance, and $y_i$ is the corresponding class such that $y_i \in \{1, ..., m\}$. In image classification a *dot-product maximization* model consists of two parts. The first is the image encoder, denoted as $f_\theta : \mathcal{X} \to \mathbb{R}^d$, which is responsible for representing the input image as a $d$-dimensional vector, $f_\theta(x) \in \mathbb{R}^d$, where $\theta$ is a set of learnable parameters. The second

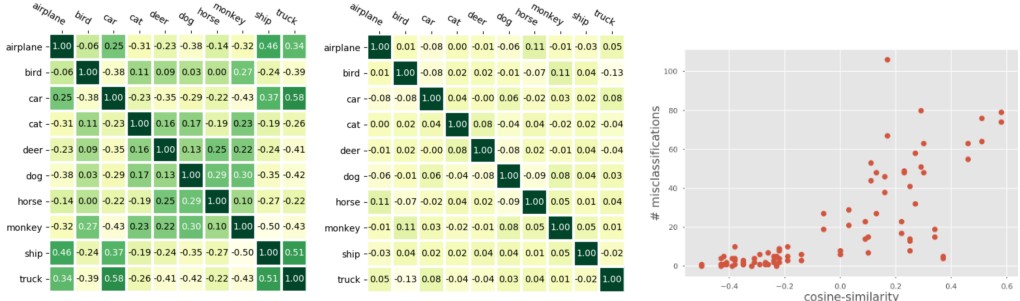

Figure 2: The matrices show the cosine-similarity between the class vectors of non-fixed (left) and fixed (middle) dot-product maximization models trained on STL-10 dataset. Right is a plot showing the number of misclassifications as a function of the cosine-similarity between class vectors.

part of the model is the classification layer, which is composed of learnable parameters denoted as $W \in \mathbb{R}^{m \times d}$. Matrix $W$ can be viewed as $m$ vectors, $w^1, \ldots, w^m$, where each vector $w^i \in \mathbb{R}^d$ can be considered as the representation vector associated with the $i$-th class. For simplicity, we omitted the bias terms and assumed they can be included in $W$.

A consideration that is taken when designing the classification layer is choosing the operation applied between the matrix $W$ and the image representation vector $f_\theta(x)$. Most commonly, a dot-product operation is used, and the resulting vector is referred to as the *logits* vector. For training the models, a softmax operation is applied over the logits vector, and the result is given to a cross-entropy loss which should be minimized. That is,

$$\underset{w^1, \ldots, w^m, \theta}{\arg \min} \sum_{i=0}^{N} - \log \frac{e^{w^{y_i} \cdot f_\theta(x_i)}}{\sum_{j=1}^{m} e^{w^j \cdot f_\theta(x_i)}} = \underset{w^1, \ldots, w^m, \theta}{\arg \min} \sum_{i=0}^{N} - \log \frac{e^{\|w^{y_i}\| \, \|f_\theta(x_i)\| \cos(\alpha_{y_i})}}{\sum_{j=1}^{m} e^{\|w^j\| \, \|f_\theta(x_i)\| \cos(\alpha_j)}}. \quad (1)$$

The equality holds since $w^{y_i} \cdot f_\theta(x_i) = \|w^{y_i}\| \|f_\theta(x_i)\| \cos(\alpha_{y_i})$, where $\alpha_k$ is the angle between the vectors $w^k$ and $f_\theta(x_i)$.

We trained three dot-product maximization models with different known CNN architectures over four datasets, varying in image size and number of classes, as described in detail in Section 4.1. Since these models optimize the dot-product between the image vector and its corresponding learnable class vectors, we refer to these models as ***non-fixed*** *dot-product maximization* models.

Inspecting the matrix $W$ of the trained models reveals that visually similar classes have their corresponding class vectors close in space. On the left panel of Fig. 2, we plot the cosine-similarity between the class vectors that were learned by the non-fixed model which was trained on the STL-10 dataset. It can be seen that the vectors representing *vehicles* are relatively close to each other, and far away from vectors representing *animals*. Furthermore, when we inspect the class vectors of non-fixed models trained on CIFAR-100 (100 classes) and Tiny ImageNet (200 classes), we find even larger similarities between vectors due to the high visual similarities between classes, such as *boy* and *girl* or *apple* and *orange*. By placing the vectors of visually similar classes close to each other, the inter-class separability is decreased. Moreover, we find a strong spearman correlation between the distance of class vectors and the number of misclassified examples. On the right panel of Fig. 2, we plot the cosine-similarity between two class vectors, $w^i$ and $w^j$, against the number of examples from category $i$ that were wrongly classified as category $j$. As shown in the figure, as the class vectors are closer in space, the number of misclassifications increases. In STL-10, CIFAR-10, CIFAR-100, and Tiny ImageNet, we find a correlation of 0.82, 0.77, 0.61, and 0.79, respectively (note that all possible class pairs were considered in the computation of the correlation). These findings reveal that as two class vectors are closer in space, the confusion between the two corresponding classes increases.

We examined whether the models benefit from the high angular similarities between the vectors. We trained the same models, but instead of learning the class vectors, we drew them randomly, normalized them ($\|w^j\| = 1$), and kept them *fixed* during training. We refer to these models as the ***fixed*** *dot-product maximization* models. Since the target vectors are initialized randomly, the

Table 1: Comparison between the classification accuracy of fixed and non-fixed dot-product maximization models.

| Dataset | Classes | PreActResnet18 | | ResNet18 | | MobileNetV2 | |
|---|---|---|---|---|---|---|---|
| | | Fixed | Non-Fixed | Fixed | Non-Fixed | Fixed | Non-Fixed |
| STL (96x96) | 10 | **79.7%** | 76.6% | **82.5%** | 78.1% | **81.0%** | 77.2% |
| CIFAR-10 (32x32) | 10 | 94.1% | **94.3%** | **94.2%** | 93.4% | **93.5%** | 93.1% |
| CIFAR-100 (32x32) | 100 | 75.2% | **75.3%** | **75.9%** | 74.9% | **74.4%** | 73.7% |
| Tiny ImageNet (64x64) | 200 | **59.1%** | 55.4% | **60.4%** | 58.9% | **59.4%** | 57.3% |

cosine-similarity between vectors is low even for visually similar classes. See the middle panel of Fig. 2. Notice that by fixing the class vectors and bias term during training, the model can minimize the loss in Eq. 1 only by optimizing the vector $f_\theta(x_i)$. It can be seen that by fixing the class vectors, the prediction is influenced mainly by the angle between $f_\theta$ and the fixed $w^{y_i}$ since the magnitude of $f_\theta(x_i)$ is multiplied with all classes and the magnitude of each class vectors is equal and set to 1. Thus, the model is forced to optimize the angle of the image vector towards its randomized class vector.

Table 1 compares the classification accuracy of models with a fixed and non-fixed classification layer. Results suggest that learning the matrix $W$ during training is not necessarily beneficial, and might reduce accuracy when the number of classes is high, or when the classes are visually close. Additionally, we empirically found that models with fixed class vectors can be trained with higher learning rate, due to space limitation we bring the results in the appendix (Table 7, Table 8, Table 9). By randomly drawing the class vectors, we ignore possible visual similarities between classes and force the models to minimize the loss by increasing the inter-class separability and encoding images from visually similar classes into vectors far in space, see Fig. 3.

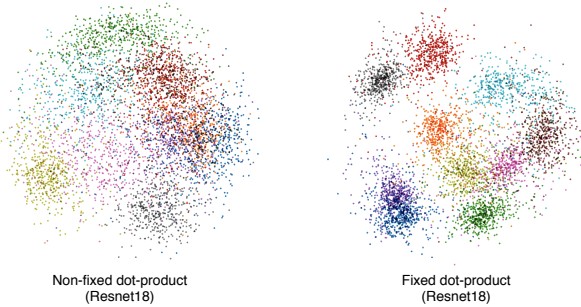

Non-fixed dot-product (Resnet18)     Fixed dot-product (Resnet18)

Figure 3: Feature distribution visualization of non-fixed and fixed dot-product maximization models trained on CIFAR-10.

## 3 FIXED COSINE-SIMILARITY MAXIMIZATION

Recently, *cosine-similarity maximization* models were proposed by Wang et al. (2017) for face verification task. The authors maximized the cosine-similarity, rather than the dot-product, between the image vector and its corresponding class vector. That is,

$$\underset{w^1,...,w^m,\theta}{\arg\min} \sum_{i=0}^{N} -\log \frac{e^{\cos(\alpha_{y_i})}}{\sum_{j=1}^{m} e^{\cos(\alpha_j)}} = \underset{w^1,...,w^m,\theta}{\arg\min} \sum_{i=0}^{N} -\log \frac{e^{\frac{w^{y_i} \cdot f_\theta(x_i)}{\|w^{y_i}\|\|f_\theta(x_i)\|}}}{\sum_{j=1}^{m} e^{\frac{w^j \cdot f_\theta(x_i)}{\|w^j\|\|f_\theta(x_i)\|}}} \quad (2)$$

Comparing the right-hand side of Eq. 2 with Eq. 1 shows that the cosine-similarity maximization model simply requires normalizing $f_\theta(x)$, and each of the class representation vectors $w^1, ..., w^m$, by dividing them with their $l_2$-norm during the forward pass. The main motivation for this reformulation is the ability to learn more discriminative features in face verification by encouraging intra-class compactness and enlarging the inter-class separability. The authors showed that dot-product maximization models learn radial feature distribution; thus, the inter-class separability and intra-class

compactness are not optimal (for more details, see the discussion in Wang et al. (2017)). However, the authors found that cosine-similarity maximization models as given in Eq. 2 fail to converge and added a scaling factor $S \in \mathbb{R}$ to multiply the logits vector as follows:

$$\arg\min_{w^1,...,w^m,\theta} \sum_{i=0}^{N} -\log \frac{e^{S \cdot \cos(\alpha_{y_i})}}{\sum_{j=1}^{m} e^{S \cdot \cos(\alpha_j)}} \quad (3)$$

This reformulation achieves improved results for face verification task, and many recent alternations also integrated the scaling factor $S$ for convergences when optimizing the cosine-similarity Wang et al. (2018b); Wojke & Bewley (2018); Deng et al. (2019); Wang et al. (2018a); Fan et al. (2019).

According to Wang et al. (2017), cosine-similarity maximization models fail to converge when $S = 1$ due to the low range of the logits vector, where each cell is bounded between $[-1, 1]$. This low range prevents the predicted probabilities from getting close to 1 during training, and as a result, the distribution over target classes is close to uniform, thus the loss will be trapped at a very high value on the training set. Intuitively, this may sound a reasonable explanation as to why directly maximizing the cosine-similarity fails to converge ($S = 1$). Note that even if an example is correctly classified and well separated, in its best scenario, it will achieve a cosine-similarity of 1 with its ground-truth class vector, while for other classes, the cosine-similarity would be $(-1)$. Thus, for a classification task with $m$ classes, the predicted probability for the example above would be:

$$P(Y = y_i | x_i) = \frac{e^1}{e^1 + (m-1) \cdot e^{-1}} \quad (4)$$

Notice that if the number of classes $m = 200$, the predicted probability of the correctly classified example would be at most 0.035, and cannot be further optimized to 1. As a result, the loss function would yield a high value for a correctly classified example, even if its image vector is placed precisely in the same direction as its ground-truth class vector.

As in the previous section, we trained the same models over the same datasets, but instead of optimizing the dot-product, we optimized the cosine-similarity by normalizing $f_\theta(x_i)$ and $w^1, ..., w^m$ at the forward pass. We denote these models as ***non-fixed*** *cosine-similarity maximization* models. Additionally, we trained the same cosine-similarity maximization models with fixed random class vectors, denoting these models as ***fixed*** *cosine-similarity maximization*. In all models (fixed and non-fixed) we set $S = 1$ to directly maximize the cosine-similarity, results are shown in Table 2.

Surprisingly, we reveal that **the low range of the logits vector *is not* the cause preventing from cosine-similarity maximization models from converging**. As can be seen in the table, fixed cosine-maximization models achieve significantly high accuracy results by up to 53% compared to non-fixed models. Moreover, it can be seen that fixed cosine-maximization models with $S = 1$ can also outperform dot-product maximization models. This finding demonstrates that while the logits are bounded between $[-1, 1]$, the models can still learn high-quality representations and decision boundaries.

Table 2: Classification accuracy of fixed and non-fixed cosine-similarity maximization models. In all models $S = 1$.

| Dataset | Classes | PreActResnet18 | | ResNet18 | | MobileNetV2 | |
|---|---|---|---|---|---|---|---|
| | | Fixed | Non-Fixed | Fixed | Non-Fixed | Fixed | Non-Fixed |
| STL (96x96) | 10 | **83.7%** | 58.9% | **83.2%** | 51.6% | **77.1%** | 54.1% |
| CIFAR-10 (32x32) | 10 | **93.3%** | 80.1% | **93.2%** | 70.4% | **92.2%** | 57.6% |
| CIFAR-100 (32x32) | 100 | **74.6%** | 29.9% | **74.4%** | 27.1% | **72.6%** | 19.7% |
| TinyImagenet (64x64) | 200 | **50.4%** | 19.9% | **47.6%** | 16.7% | **41.6%** | 13.2% |

We further investigated the effects of $S$ and train for comparison the same fixed and non-fixed models, but this time we used grid-search for the best performing $S$ value. As can be seen in Table 3, increasing the scaling factor $S$ allows non-fixed models to achieve higher accuracies over all datasets. Yet, there is no benefit at learning the class representation vectors instead of randomly drawing them and fixing them during training when considering models' accuracies.

To better understand the cause which prevents non-fixed cosine-maximization models from converging when $S = 1$, we compared these models with the same models trained by setting the optimal

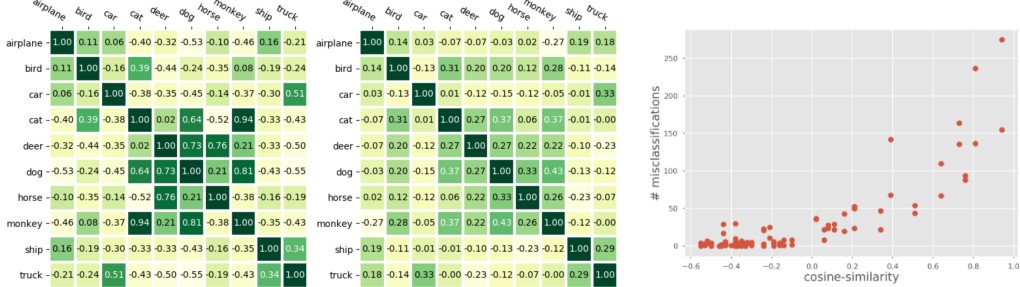

Figure 4: The matrices show the cosine-similarity between the class vectors of non-fixed cosine-similarity maximization models, trained on STL-10 dataset, with $S = 1$ (left), and $S = 20$ (middle). Right is a plot showing the relationship between the number of misclassification as a function of the cosine-similarity between class vectors.

Table 3: Comparison between the classification accuracy of fixed and non-fixed cosine-similarity maximization models with their optimal $S$

| Dataset | Classes | PreActResnet18 | | ResNet18 | | MobileNetV2 | |
|---------|---------|-------|-----------|-------|-----------|-------|-----------|
| | | Fixed | Non-Fixed | Fixed | Non-Fixed | Fixed | Non-Fixed |
| STL (96x96) | 10 | **83.8%** | 79.0% | **83.9%** | 79.4% | **82.4%** | 81.5% |
| CIFAR-10 (32x32) | 10 | **93.5%** | **93.5%** | **93.3%** | 93.0% | 92.6% | **92.8%** |
| CIFAR-100 (32x32) | 100 | 74.6% | **74.8%** | **74.6%** | 73.4% | **73.7%** | 72.6% |
| TinyImagenet (64x64) | 200 | 53.8% | **54.5%** | 54.3% | **55.6%** | **53.9%** | 53.5% |

$S$ scalar. For each model we measured the distance between its learned class vectors and compared these distances to demonstrate the effect of $S$ on them. Interestingly, we found that as $S$ increased, the cosine-similarity between the class vectors decreased. Meaning that by increasing $S$ the class vectors are further apart from each other. Compare, for example, the left and middle panels in Fig. 4, which show the cosine-similarity between the class vectors of models trained on STL with $S = 1$ and $S = 20$, respectively.

On the right panel in Fig. 4, we plot the number of misclassification as a function of the cosine-similarity between the class vectors of the non-fixed cosine-maximization model trained on STL-10 with $S = 1$. It can be seen that the confusion between classes is high when they have low angular distance between them. As in previous section, we observed strong correlations between the closeness of the class vectors and the number of misclassification. We found correlations of 0.85, 0.87, 0.81, and 0.83 in models trained on STL-10, CIFAR-10, CIFAR-100, and Tiny ImageNet, respectively.

By integrating the scaling factor $S$ in Eq. 4 we get

$$P(Y = y_i | x_i) = \frac{e^{S \cdot 1}}{e^{S \cdot 1} + (m - 1) \cdot e^{S \cdot (-1)}} \tag{5}$$

Note that by increasing $S$, the predicted probability in Eq. 5 increases. This is true even when the cosine-similarity between $f(x_i)$ and $w^{y_i}$ is less than 1. When $S$ is set to a large value, the gap between the logits increases, and the predicted probability after the softmax is closer to 1. As a result, **the model is discouraged from optimizing the cosine-similarity between the image representation and its ground-truth class vector to be close to 1, since the loss is closer to 0**. In Table 4, we show that as we increase $S$, the cosine-similarity between the image vectors and their predicted class vectors decreases.

These observations can provide an explanation as to why non-fixed models with $S = 1$ fail to converge. By setting $S$ to a large scalar, the image vectors are spread around their class vectors with a larger degree, preventing the class vectors from getting close to each other. As a result, the inter-class separability increases and the misclassification rate between visually similar classes decreases. In contrast, setting $S = 1$ allows models to place the class vectors of visually similar classes closer in space and leads to a high number of misclassification. However, **a disadvantage of increasing $S$ and setting it to a large number is that the intra-class compactness is violated since image**

**vectors from the same class are spread and encoded relatively far from each other**, see Fig. 5. Fixed cosine-maximization models successfully converge when $S = 1$, since the class vectors are

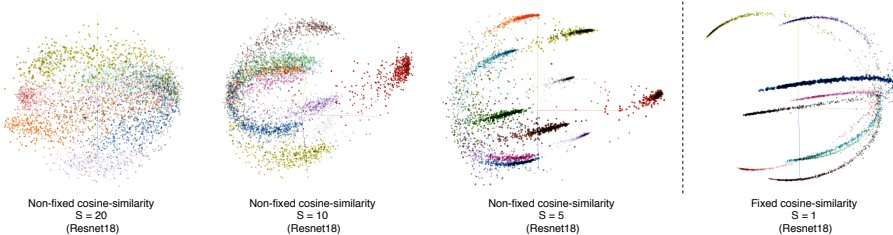

Non-fixed cosine-similarity
S = 20
(Resnet18)

Non-fixed cosine-similarity
S = 10
(Resnet18)

Non-fixed cosine-similarity
S = 5
(Resnet18)

Fixed cosine-similarity
S = 1
(Resnet18)

Figure 5: Feature distribution visualization of fixed and non-fixed cosine-maximization Resnet18 models trained on CIFAR-10.

initially far in space from each other. By randomly drawing the class vectors, models are required to encode images from visually similar classes into vectors, which are far in space; therefore, the inter-class separability is high. Additionally, the intra-class compactness is improved since models are encouraged to maximize the cosine-similarity to 1 as $S$ can be set to 1, and place image vectors from the same class close to their class vector. We validated this empirically by measuring the average cosine-similarity between image vectors and their predicted classes' vectors in fixed cosine-maximization models with $S = 1$. We obtained an average cosine-similarity of roughly 0.95 in all experiments, meaning that images from the same class were encoded compactly near their class vectors.

In conclusion, although non-fixed cosine-similarity maximization models were proposed to improve the caveats of dot-product maximization by improving the inter-class separability and intra-class compactness, their performance are significantly low without the integration of a scaling factor to multiply the logits vector. Integrating the scaling factor and setting it to $S > 1$ decrease intra-class compactness and introduce a trade-off between accuracy and intra-class compactness. By fixing the class vectors, cosine-similarity maximization models can have both high performance and improved intra-class compactness. Meaning that multiple previous works (Wang et al. (2018b); Wojke & Bewley (2018); Deng et al. (2019); Wang et al. (2018a); Fan et al. (2019)) that adopted the cosine-maximization method and integrated a scaling factor for convergence, might benefit from improved results by fixing the class vectors.

Table 4: Average cosine-similarity results between image vectors and their predicted class vectors, when $S$ is set to 1, 20, and 40. Results are from non-fixed cosine-similarity maximization models trained on CIFAR-10 (C-10), CIFAR-100 (C-100), STL, and Tiny ImageNet (TI).

| S=1 | | | | S=20 | | | | S=40 | | | |
|------|-------|------|------|------|-------|------|------|------|-------|------|------|
| C-10 | C-100 | STL | TI | C-10 | C-100 | STL | TI | C-10 | C-100 | STL | TI |
| 0.99 | 0.99 | 0.97 | 0.98 | 0.88 | 0.63 | 0.77 | 0.62 | 0.36 | 0.53 | 0.21 | 0.37 |

## 4    GENERALIZATION AND ROBUSTNESS TO CORRUPTIONS

In this section we explore the generalization of the evaluated models to the learned concepts and measure their robustness to image corruptions. We do not aim to set a state-of-the-art results but rather validate that by fixing the class vectors of a model, the model's generalization ability and robustness to corruptions remain competitive.

### 4.1    TRAINING PROCEDURE

To evaluate the impact of ignoring the visual similarities in the classification layer we evaluated the models on CIFAR-10, CIFAR-100 Krizhevsky et al. (2009), STL Coates et al. (2011), and Tiny ImageNet[1] (containing 10, 100, 10, and 200 classes, respectively). For each dataset, we trained Resnet18 He et al. (2016a), PreActResnet18 He et al. (2016b), and MobileNetV2 Sandler et al. (2018) models

---

[1]https://tiny-imagenet.herokuapp.com/

Table 5: Classification accuracy of fixed and non-fixed models for the generalization sets.

| Training set | Evaluating set | Dot-product | | Cosine-similarity | |
|---|---|---|---|---|---|
| | | Fixed | Non-Fixed | Fixed | Non-Fixed |
| STL | STL Gen | 53.7% | 50.9% | **54.1%** | 50.4% |
| CIFAR-10 | CIFAR-10.1 | **87.1%** | 86.9% | 85.7% | 85.9% |
| CIFAR-100 | CIFAR-100 Gen | 34.8% | 32.9% | 35.1% | **35.4%** |

with fixed and non-fixed class vectors. All models were trained using stochastic gradient descent with momentum. We used the standard normalization and data augmentation techniques. Due to space limitations, the values of the hyperparameters used for training the models can be found under our code repository. We normalized the randomly drawn, fixed class representation vectors by dividing them with their $l_2$-norm. All reported results are the averaged results of 3 runs.

## 4.2 GENERALIZATION

For measuring how well the models were able to generalize to the learned concepts, we evaluated them on images containing objects from the same target classes appearing in their training dataset. For evaluating the models trained on STL-10 and CIFAR-100, we manually collected 2000 and 6000 images ,respectively, from the publicly available dataset *Open Images V4* Krasin et al. (2017). For CIFAR-10 we used the CIFAR-10.1 dataset Recht et al. (2018). All collected sets contain an equal number of images for each class. We omitted models trained on Tiny ImageNet from the evaluation since we were not able to collect images for all classes appearing in this set. Table 5 summarizes the results for all the models. Results suggest that excluding the class representation vectors from training, does not decrease the generalization to learned concepts.

## 4.3 ROBUSTNESS TO CORRUPTIONS

Next, we verified that excluding the class vectors from training did not decrease the model's robustness to image corruptions. For this we apply three types of algorithmically generated corruptions on the test set and evaluate the accuracy of the models on these sets. The corruptions we apply are impulse-noise, JPEG compression, and de-focus blur. Corruptions are generated using Jung (2018), and available under our repository. Results, as shown in Table 6, suggest that randomly drawn fixed class vectors allow models to be highly robust to image corruptions.

Table 6: Classification accuracy of fixed and non-fixed models on corrupted test set's images.

| Corruption type | Test set | Dot-product | | Cosine-similarity | |
|---|---|---|---|---|---|
| | | Fixed | Non-fixed | Fixed | Non-fixed |
| Salt-and-pepper | STL | 52.9% | 49.4% | **57.2%** | 44.9% |
| | CIFAR-10 | 52.6% | 49.5% | 49.8% | **52.9%** |
| | CIFAR-100 | 36.1% | 36.6% | **41.3%** | 40.9% |
| | Tiny ImageNet | **30.8%** | 26.1% | 28.6% | 27.9% |
| JPEG compression | STL | **78.1%** | 75.8% | 76.1% | 73.3% |
| | CIFAR-10 | 71.8% | 71.5% | 71.4% | **72.4%** |
| | CIFAR-100 | 43.8% | 42.3% | **44.4%** | 43.3% |
| | Tiny ImageNet | **38.8%** | 33.0% | 35.1% | 34.4% |
| Blurring | STL | 37.7% | 35.0% | **39.4%** | 36.1% |
| | CIFAR-10 | 41.3% | 41.1% | **41.9%** | 41.1% |
| | CIFAR-100 | **24.8%** | 24.4% | 22.4% | 23.1% |
| | Tiny ImageNet | **16.4%** | 13.4% | 12.9% | 12.4% |

## 5 CONCLUSION

In this paper, we propose randomly drawing the parameters of the classification layer and excluding them from training. We showed that by this, the inter-class separability, intra-class compactness, and the overall accuracy of the model can improve when maximizing the dot-product or the cosine similarity between the image representation and the class vectors. We analyzed the cause that prevents the non-fixed cosine-maximization models from converging. We also presented the generalization abilities of the fixed and not-fixed classification layer.

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

# A VARIOUS LEARNING RATE INITIALIZATIONS

Table 7: Comparison between the classification accuracy of fixed and non-fixed PreActResnet18 models using various LR initializations. Nan values indicate that the model failed to converge.

| Dataset | LR = 0.1 | | LR = 0.01 | | LR = 0.001 | | LR = 0.0001 | |
| --- | --- | --- | --- | --- | --- | --- | --- | --- |
| | Fixed | Non-Fixed | Fixed | Non-Fixed | Fixed | Non-Fixed | Fixed | Non-Fixed |
| STL (96x96) | **79.4**% | 76.6 | **79.7**% | 75.9% | **78.2**% | 76.5% | **70.6**% | 68.9% |
| CIFAR-100 (32x32) | **72.2**% | NaN | 75.2% | **75.3**% | **74.4**% | 72.5% | – | – |
| TinyImagenet (64x64) | **59.1**% | NaN | **55.6**% | 55.4% | **52.9**% | 52.1% | – | – |

Table 8: Comparison between the classification accuracy of fixed and non-fixed Resnet18 models using various LR initializations.

| Dataset | LR = 0.1 | | LR = 0.01 | | LR = 0.001 | | LR = 0.0001 | |
| --- | --- | --- | --- | --- | --- | --- | --- | --- |
| | Fixed | Non-Fixed | Fixed | Non-Fixed | Fixed | Non-Fixed | Fixed | Non-Fixed |
| STL (96x96) | **82.5**% | 75.9% | **81.2**% | 78.1% | **79.1**% | 76.5% | **77.9**% | 75.8% |
| CIFAR-100 (32x32) | **74.8**% | 73.1% | **75.9**% | 74.4% | **74.5**% | 72.6% | – | – |
| TinyImagenet (64x64) | **60.1**% | 59.0% | **58.2**% | 57.1% | **54.4**% | 53.9% | – | – |

Table 9: Comparison between the classification accuracy of fixed and non-fixed MobileNetV2 models using various LR initializations.

| Dataset | LR = 0.1 | | LR = 0.01 | | LR = 0.001 | | LR = 0.0001 | |
| --- | --- | --- | --- | --- | --- | --- | --- | --- |
| | Fixed | Non-Fixed | Fixed | Non-Fixed | Fixed | Non-Fixed | Fixed | Non-Fixed |
| STL (96x96) | **80.8**% | 75.9% | **81.0**% | 76.1% | **74.4**% | 74.4% | 73.1% | **73.9**% |
| CIFAR-100 (32x32) | **67.5**% | 64.2% | **75.1**% | 73.8% | **70.8**% | 70.4% | – | – |
| TinyImagenet (64x64) | **56.8**% | 55.1% | **59.3**% | 57.1% | **51.2**% | 49.9% | – | – |

