# OpenReview forum: "Redesigning the Classification Layer by Randomizing the Class Representation Vectors"
_ICLR.cc/2021/Conference — Reject_

### Official Review · AnonReviewer3 · 2020-10-25
**The model is reasonable while the experiments are not convincing**

**Rating:** 5
**Confidence:** 4

**Review:**

This paper proposed a classification layer by randomizing the class representation vectors. This paper first analyses the class vector distributions between different training strategies, and then proposed the randomized class vector to improve the representation learning performance. The proposed model is further extended and analyzed for the fixed cosine-similarity maximization setting. The experiments demonstrate the effectiveness of the proposed method compared with the basic/vanilla baselines.

Pros:
The motivation of this paper is comprehensive. Some quantitative and visual experimental results introduced the motivation of the proposed model. The randomization weights also provide a novel view for solving more machine learning problems. This is a good point.

Cons:
My main concern is the experimental results. The experiments are mainly done for evaluating fixed and non-fixed models without any other state-of-the-art methods, and the exact performance is considerably low compared with other state-of-the-art methods.
To this end, it is hard to confirm the effectiveness of the model. For example, 1) even though the visualization results are good (Figure 3), it does not mean the final performance is better. 2) the original model could be over-fitting, and a random and fixed weight layer could be considered as a regularizer. There are some experiments should be done: 1) Compare this method with relevant methods such as NormFace and ArcFace to proof the effectiveness of this approach. 2) Compared with exact performance in face relevant datasets and compared with other SOTA methods.

---

> ### Author Response · Authors · 2020-11-16
> **answer**
>
> We thank the reviewer for his detailed feedback on our paper. We hope to answer his questions below.
>
> Our main goal of this paper is not to introduce SOTA results but rather to show the interesting phenomena of improving model performance (in terms of accuracy and intra-class compactness, and inter-class separation) by invalidating the visual similarities from the target vectors. Therefore we trained non-fixed models on known datasets and compared the same exact models with fixed target vectors.
>
> Regarding the visualization in figure-3, the purpose of this visualization (and the visualization in figure-5) is to show the improved intra-class compactness and inter-class separability (besides, we provide the distances between the examples and the target vectors in table-4 to highlight the improvements). To show that the final performance is improved, we bring table-1, table-2, table-3, table-7, table-8, and table-9, where we mention the accuracy achieved by the models. To demonstrate that the models are not simply overfitting, we added the generalization and robustness to noise experiments (table-5 and table-6) that demonstrate the fact that the fixed and non-fixed models are not overfitting to the in-distribution datasets.
>
> Regarding the comparison with NormFace, please note that in section-3, all the comparisons are against non-fixed NormFace models (we state this at the beginning of the section). We will highlight this in the paper.

---

### Official Review · AnonReviewer1 · 2020-10-27
**Training a multi-class image classification model by fixing the weights of the classification layer**

**Rating:** 4
**Confidence:** 4

**Review:**

**Summary:**

This paper introduces a new approach to learn a multi-class image classification model by fixing the weights of the classification layer. The authors propose to draw the class vectors randomly and set them as fixed during training instead of training them. They analyze this approach when a model is trained with a categorical cross-entropy and or softmax-cosine loss. The proposed approach is tested on 4 datasets: STL, CIFAR-10, CIFAR-100, TinyImagenet

**Reasons for score:**

I do not think the technical contribution is strong enough for ICLR. The idea is interesting but the empirical validation of the idea should be improved and some claims should be proved.


**Pros:**

- The idea of using fixed-representation is interesting. It can help to reduce the training time.
- The authors explain why cosine-similarity maximization models cannot converge to 0.

**Cons:**

The title looks very interesting: “Redesigning the Classification Layer by Randomizing the Class Representation Vectors”. But after reading the paper, it is only about mullti-class image classification. There is no study about other types of data or the multi-label setting. The authors should use a title more accurate about the content of their paper.

Overall, the structure of the paper should be improved. It is quite difficult to read because several sections are a mix of model contributions and experimental results. Maybe using subsections can help to separate the model contributions and experimental results. Also, some information is not at the right place and some sections should be reorganized. For example, the datasets and models are presented in section 4.1 but some results are presented in section 2. The authors should also add a related work section to clearly state the motivations and explain the difference with other approaches.

The authors proposed to randomly initialize the weights of the classification layer but they do not clearly explain how the weights are initialized. There are several standard approaches to initialize weights like uniform, normal, Xavier uniform, Xavier normal, Kaiming uniform, Kaiming normal. It can improve the paper if the authors compare these initialization mechanisms. Similarly, the authors should analyze the results for several runs to see how the fixed weights approach is sensitive to the random initialization.

I have a conceptual problem with fixing the bias. The bias is sampled so it means it can have a large or small value. Let’s take an example with 2 classes. The class A can have a large bias (e.g. 0.5) but other class B can have a small value (e.g. -0.5). It means that the class B has a negative bias and will usually have lower scores than A just because there is a difference of 1 between these biases. I am not sure that it is a good idea and there is no motivation about that in the paper. The authors should analyze the bias initialization because it is important.

It is important to show the variance when the model is evaluated on several runs (section 4). It can help to understand how the model is sensible to the initialization.

It is well known that the SGD is sensible to its hyper-parameter and in particular the learning rate. The model will not converge if the learning rate is too large or too small. The authors should explain how they choose the hyper-parameters. I also wonder how the results are specific to the optimizer. Are the conclusions of the analysis the same for other popular optimizers like Adam.

“These observations can provide an explanation as to why non-fixed models with S = 1 fail to converge.” (page 6): For me it explains why the model cannot converge to 0 but it does not explain why the model fails to converge. They are two different problems.

In the abstract and in some other parts of the paper the authors claim they improve the compactness of the model. But they never show it. They did not define how they measure the compactness of a model. They should clearly present the definition of compactness, and what approach they used to compute it. Based on my knowledge, measuring the compactness of a model is not easy.

The authors should results on low resolution dataset (less than 100*100). I wonder if the results can be generalized to larger resolution dataset. For example, does it also work on ImageNet that has more images, larger resolution images and more classes (1000). I also wonder if it works on other type of datasets like fine-grained datasets (e.g. CUB-200, Stanford Cars, FGVC Aircraft). Also, how does it adapt to new domains like medical images and natural scenes.

I am not convinced that ignoring the visual similarities between classes is a good idea. I think it is important to build spaces that encode some semantic structure. For example, I think it is important to encode that two bird species are more semantically similar than a bird and a car.

It is not clear why the authors decided to focus on the cosine-similarity maximization models. They should motivate this decision more because these models are not so popular.

The authors claimed that “the low range of the logits vector is not the cause preventing from cosine-similarity maximization models from converging” (page 5) but they did not show results to prove it. The authors should analyze the range of the logits. The current analysis does not allow us to understand if it is because of the range of value, or the normalization of the weights or a bad tuning of some hyper-parameters.

**Minor comments:**

The authors should give more information on how they generated the figures 3 and 5.

---

> ### Author Response · Authors · 2020-11-16
> **answer**
>
> We thank the reviewer for his detailed feedback on our paper. We hope to answer his questions below.
>
> 1) Reg. the initialization of the classification layer’s weights - we have tested multiple initializations and observed that the main factor which influences the model’s performance is the distance between the target vectors. Therefore, we mentioned in the paper that we draw the vectors randomly (each cell in the vector is drawn randomly between [-100,100]) and normalize them. This can be also be seen in the attached code. We will make it clearer in the paper.
>
> 2) Reg. fixing the bias term - we tried several strategies where we 1) omit the bias term 2) do not fix the bias term 3) initialize it with small/large values. In all cases, the results were similar, and therefore we have not reported it. Ignoring the bias term is done in the past in multiple works (for example, in “NormFace: L2 Hypersphere Embedding for Face Verification” - last line in section 3.1). Our observations are also in line with “FIX YOUR CLASSIFIER: THE MARGINAL VALUE OF TRAINING THE LAST WEIGHT LAYER” - which also proposes fixing the bias term.
>
> 3) Reg. the claim - “These observations can provide an explanation as to why non-fixed models with S = 1 fail to converge.” (page 6)”.  Several papers claimed that the bounded range of the logits causes converages issues, please see: 1) Sec 2.2 in “Fix your classifier: the marginal value of training the last weight layer”    2) Sec 3.3 in “NormFace: L2 Hypersphere Embedding for Face Verification”.  In our work we wanted to shade a light and demonstrate that models can converge even with the limited range of the logits. We invalidate this claim by showing the when we fix the target vectors the model can converge even though the logits are bounded between [-1,1]. Afterwards, as we demonstrated that the range is not the cause which prevent from non-fixed models to converaged we moved to investigate what happens to non-fixed models when we set large S value (as we demonstrate in table-2 and table-3 the only difference between the non-fixed models is the value of S). We showed that S is set to large value, the target vectors are placed far from each other compared to the case where S is set to 1 (in figure 4 + figure 5), and therefore we hypothesize that the convergence issues are due to the distance between the target vectors and not due to the bounded range of the logits.
>
> 4) Reg. the claim that “the low range of the logits vector is not the cause preventing cosine-similarity maximization models from converging” - as we mentioned in the previous bullet (and in the paper in section 3), the claim that the models does not converge due to the low range of the logits was published before in multiple prior works. We showed that the low range is not the cause preventing models from converging by showing that the models are able to converge and achieve high accuracy with S=1 (meaning that the logits are bounded between [-1,1]). The models failed to converge not because of bad hyperparameters tuning or weight normalization (as this problem was reported in multiple works)
>
> 5) Reg. the reason for focusing on the cosine-similarity maximization models - in ICLR 2018 the paper “Fix your classifier: the marginal value of training the last weight layer” proposed to fix the classification layer. They compared fixed and non-fixed models and showed that there is no benefit in training the last layer. However, as we mentioned in the introduction, the authors of the paper compared non-fixed dot-product models against fixed cosine-similarity models without the awareness that they are not equivalent. We further investigated and showed that when a fair comparison is made (fixed dot-product vs non-fixed dot-product and fixed cosine-similarity vs non-fixed cosine-similarity), the fixed models outperform with a significant gap and not only achieve comparable results. Moreover, as mentioned in section-3 multiple works adopted and integrated the scaling factor to allow convergence while not knowing the caveats of adding the scaling factor, and therefore we investigated the effects of fixing the classification layer in cosine-similarity models.
>
> 6) As for the datasets we use in our work, we followed the datasets used by “Fix your classifier: the marginal value of training the last weight layer”, which trained on Cifar10 (1 model), Cifar100 (1 model), and Imagenet (3 models). In our work, we trained on Cifar10 (3 models), Cifar100 (3 models), STL (3 models), and TinyImagenet (3 models). We have not trained on ImageNet due to a lack of computational resources.

---

### Official Review · AnonReviewer4 · 2020-10-28
**Interesting observation but limited demonstrated scope**

**Rating:** 5
**Confidence:** 4

**Review:**

The paper explores deeper into the specific classification layer of a standard supervised learning system. The core idea of the paper is to randomly initialize and then fix the classification layer weights and train the network leading improved discrimination.
The writing is satisfactory and the paper develops the ideas sufficiently well to help any reader who is a beginner in this area.

One of the major concerns regarding the work is that it seems to have is the relatively limited amount of contribution given the context of the current venue. This is no doubt an interesting phenomenon, however, previous works investigating cosine similarity losses have tested their approaches on much larger problems such as large scale face recognition and full Imagenet. The paper currently derives its intuitions from object recognition problems which have very different behavior than problems like face recognition where the number of classes is large yet the number of samples per class is much lower.

That said, given the limited scale of the experiments, the paper does offer a wider variety of results supporting its claims. Nonetheless, given the simplicity of the idea, the paper fails to push envelope of results on any of these datasets. Lastly, the performance gains in Table 3 seem limited, given that only one run was performed for each dataset.

---

> ### Author Response · Authors · 2020-11-17
> **answer**
>
> We thank the reviewer for his detailed feedback on our paper. We hope to address his concerns below.
>
> Reg. the reason for focusing on the cosine-similarity maximization models - in ICLR 2018 the paper “Fix your classifier: the marginal value of training the last weight layer” proposed to fix the classification layer. They compared fixed and non-fixed models and showed that there is no benefit in training the last layer. However, as we mentioned in the introduction, the authors of the paper compared non-fixed dot-product models against fixed cosine-similarity models without the awareness that they are not equivalent. We further investigated and showed that when a fair comparison is made (fixed dot-product vs non-fixed dot-product and fixed cosine-similarity vs non-fixed cosine-similarity), the fixed dot-product models can outperform with a significant gap and not only achieve comparable results. As for the datasets we use in our work, we followed the datasets used by “Fix your classifier: the marginal value of training the last weight layer”, which trained on Cifar10 (1 model), Cifar100 (1 model), and Imagenet (3 models). In our work, we trained on Cifar10 (3 models), Cifar100 (3 models), STL (3 models), and TinyImagenet (3 models). We have not trained on ImageNet due to a lack of computational resources.
>
> As for table-3, please notice that cosine-similarity models with fixed classification layer not only achieve slightly improved results but as we demonstrate through section-3 they benefit from improved intra-class compactness and inter-class separability, does not require tuning the scaling parameter (since it is not needed), and have improved the robustness to noise (please see table-6 where the fixed cosine-similarity models outperform non-fixed model on 9/12 tested cases).
>
> Cosine-similarity maximization models are only 1 part out of 2 approaches that we have tested (the second is the dot-product maximization models). It was interesting in our opinion to report the observation that although the scaling factor is widely adopted and is extensively used in classification models, it brings several caveats that can be addressed by fixing the classification layer.
>
> Lastly, please notice that in section 4.1 we mentioned that the reported results are the average of 3 runs (for each dataset, we ran the same model 3 times); we will add the confidence intervals to the appendix section.

---

### Official Review · AnonReviewer2 · 2020-10-29
**A provocative idea begging for more evidence**

**Rating:** 4
**Confidence:** 4

**Review:**

This paper claims that in the context of multi-class image classification using neural networks, in the final classification layer, if we use randomly initialized parameters (with normalization) without any training, we can achieve better performance than if we train those parameters. This is an intriguing claim that can potentially have a very broad impact. The authors provide some motivations based on the error-similarity plots, but no theoretical backing. Without convincing theoretical support, such a claim can only be established through extensive and rigorous experimentation, and I find the experiment description in this paper is short on delivering strong evidence. For example, how many runs to achieve the results in Tables 1-3? What are confidence intervals on the results?  Any statistical significance test done? How were hyperparameters selected?  What about the performance on the ImageNet dataset, which has more classes than the datasets reported in the paper? What distribution was used to initialize the random weights in the classification layer? Is the performance sensitive to the distribution? Is the performance sensitive to the complexity of the model used to learn the representation? How does this compare to other ways of improve multi-class classification such as softmax temperature annealing, label smoothing, adding regularization, etc.? Or as a stretch, does this claim generalize to problems with categorical features?

Details:
1. page 2, line 9: do you mean "maximizing the cosine-similarity"?

---

> ### Author Response · Authors · 2020-11-19
> **answer**
>
> We thank the reviewer for his detailed feedback on our paper. We hope to answer his questions below.
>
> Reg. the results in tables 1-3 (and tables 7-9) - we trained each model 3 times with random weight initializations and averaged the test set results (we have also mentioned it in section 4.1). We will add the variance of the runs to the paper. We tested multiple hyperparameters such as learning rate (which was the most influencing, thus reported the results of various runs in tables 7-9), different optimizers, with and without weight-decay, different batch sizes. The optimal parameters are included in our repository (attached to the submission).
>
> Reg. training on ImageNet dataset. Due to limited computational resources, we have not trained on this dataset. However, we trained the models over TinyImageNet, which has 200 classes.
>
> Reg. the initializations of the target vectors. We initialized the target vectors by drawing each cell a value in the range [-100, 100] uniformly and normalized the vector into a unit vector by dividing them with by the l_2 norm (can be seen in our repository - will include it in the paper). We tested several initializations (such as initializing them with sparse orthogonal vectors) and did not observe any significant change in results. The most important aspect we found is to have the vectors orthogonal to each other, and we achieve this by the randomized initialization (as can be seen in figure-2 (middle)).
>
> We have not compared our method with softmax temperature annealing/label smoothing/adding regularization, since we are not claiming to outperform any of these methods but rather to show this interesting phenomenon and highlight the boost in performance that can be achieved by fixing the classification layer while reducing the number of parameters. Moreover, fixing the classification layer can be combined with all methods mentioned above. It is interesting to investigate the benefits of these methods when fixing the classification layer; however, we feel that it is out of this paper's scope.
>
> Reg. page 2, line 9 - yes, it should be "maximizing" instead of "minimizing" - thanks for pointing this out; we will fix it.

---

### Decision · Program_Chairs · 2021-01-07
**Final Decision**

**Decision:**

Reject

**Comment:**

The reviewers are in consensus that this paper is not ready for publication: cited concerns include simple (interesting) ideas but need to be carefully analyzed empirically, contextualized (other similar studies exist), identifying convincing empirical evidences,. etc.


The AC recommends Reject.